# Analysis of polarimetric satellite measurements suggests stronger cooling due to aerosol-cloud interactions

Otto P. Hasekamp [1]*, Edward Gryspeerdt [2] & Johannes Quaas [3]

Anthropogenic aerosol emissions lead to an increase in the amount of cloud condensation nuclei and consequently an increase in cloud droplet number concentration and cloud albedo. The corresponding negative radiative forcing due to aerosol cloud interactions ($RF_{aci}$) is one of the most uncertain radiative forcing terms as reported in the 5th Assessment Report of the Intergovernmental Panel on Climate Change (IPCC). Here we show that previous observation-based studies underestimate aerosol-cloud interactions because they used measurements of aerosol optical properties that are not directly related to cloud formation and are hampered by measurement uncertainties. We have overcome this problem by the use of new polarimetric satellite retrievals of the relevant aerosol properties (aerosol number, size, shape). The resulting estimate of $RF_{aci} = -1.14$ Wm$^{-2}$ (range between $-0.84$ and $-1.72$ Wm$^{-2}$) is more than a factor 2 stronger than the IPCC estimate that includes also other aerosol induced changes in cloud properties.

[1] SRON Netherlands Institute for Space Research, Sorbonnelaan 2, 3584 CA Utrecht, The Netherlands. [2] Space and Atmospheric Physics Group, Imperial College London, London SW7 2AZ, UK. [3] Universität Leipzig, Institute for Meteorology, Stephanstr. 3, D-04103 Leipzig, Germany. *email: O.Hasekamp@sron.nl

The effect of aerosols on cloud albedo, through an increase in cloud droplet number concentration ($N_d$)[1], remains to be one of the most uncertain components of the anthropogenic radiative forcing[2]. Relationships between aerosol amount and $N_d$, observed by satellites provide an important constraint for climate models to compute the radiative forcing due to aerosol–cloud interactions. The slope of this relationship on a log-log scale is often referred to as susceptibility. Estimates of susceptibility have so far mostly been based on measurements of the aerosol optical depth (AOD)[3,4] or aerosol index (AI)[5–8]. AOD is a poor proxy for Cloud Condensation Nuclei (CCN) concentration[8–10], because it is not only affected by the concentration of CCN particles, but also depends strongly on the aerosol size (i.e., a small number of particles with large size can have the same AOD as a large number of particles with small size). Also, hydrophobic mineral dust aerosols contribute substantially to the AOD but are not very effective as CCN. This leads to an estimated susceptibility that is too weak[9]. Indeed, most models predict susceptibilities that are much higher than susceptibilities based on measurements of AOD[8,11]. An empirical way to suppress the effect of size variation on the AOD and to suppress the effect of mineral dust is to use the AI which is the product of AOD and Angstrom Exponent (a measure of the AOD spectral dependence). Although the physical meaning of the AI is only qualitative, model studies suggest[8,9] that AI is better suited than AOD to estimate RF$_{aci}$. However, studies using AI also find susceptibilities that are substantially smaller than what models predict[6]. In addition to the use of nonoptimal CCN proxies, also measurement uncertainties, especially at low aerosol concentrations, lead to an underestimate of susceptibility[12].

In this study, we use information on aerosol number, size, and shape retrieved from satellite based polarization measurements, to obtain an improved estimate of susceptibility and RF$_{aci}$. These measurements indicate that susceptibility depends strongly on aerosol size and shape. Based on this, we define a CCN proxy as the column number of particles with radius >0.15 μm, for scenes where the percentage of spherical aerosols >90%. Also, we exclude observations at small aerosol concentrations where measurement uncertainties have large effect on the derived susceptibility[12]. Using the new CCN proxy, we find susceptibilities that are substantially larger, and an RF$_{aci}$ estimate that is substantially more negative, than estimates based on AOD or AI, as used in previous studies.

## Results

**Dependence of susceptibility on size and shape.** With recent advances in aerosol retrievals from polarization measurements[13], satellite data products have become available, such as column number ($N_a$), size distribution, and particles shape. We investigate the ability of aerosols to act as CCN using POLarization and Directionality of Earth's Reflectance-3 (POLDER-3) retrieved fine and coarse mode $N_a$, effective radius ($r_{eff}$), effective variance ($v_{eff}$), and fraction of spherical particles ($f_{sp}$) from the SRON algorithm[16–18], in combination with MODerate resolution imaging spectroradiometer (MODIS) retrievals of cloud droplet effective radius and cloud optical thickness[19], from which $N_d$ is derived[11]. Following the suggestion of Dusek at al.[14], we define a CCN proxy $N_{ccn}$ as the column number of aerosol particles (in cm$^{-2}$) with radius > $r_{lim}$, where $r_{lim}$ is a threshold radius to be determined. As a proxy for particle hygroscopicity and to exclude hydrophobic mineral dust, grid cells with $f_{sp} > f_{sp,min}$ are selected, where $f_{sp,min}$, is a threshold to be determined. Using collocated POLDER-3 $N_{ccn}$ and MODIS $N_d$ retrievals on a 1° by 1° latitude–longitude grid, we determine the susceptibility $S =$

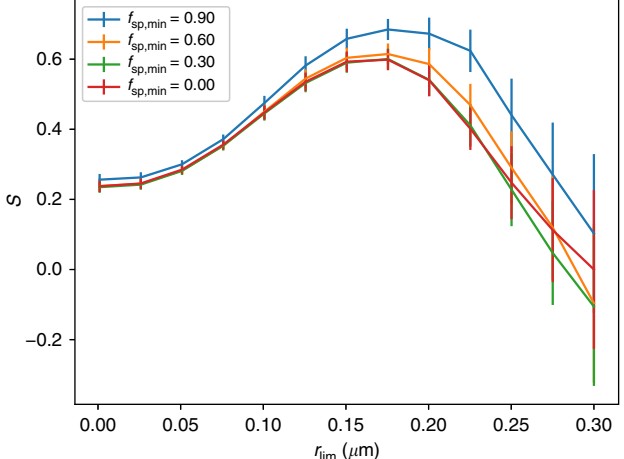

**Fig. 1** Dependence of susceptibility on size and shape. Susceptibility $S$ as function of minimum particle radius $r_{lim}$ for different values of the minimum fraction of spherical particles ($f_{sp,min}$). Error bars indicate the standard error ($2\sigma$) of the linear regression.

$\frac{d\log N_d}{d\log N_{ccn}}$ as a linear regression coefficient for binned points of $\log N_d$ versus $\log N_{ccn}$. When determining $S$, we leave out values of $N_{ccn} < 10^7$ cm$^{-2}$, because our simulator results (see Methods) indicate that inclusion of these small values may lead to a strong underestimate of $S$ because of measurement uncertainties.

Figure 1 shows $S$ for global ocean data as a function of $r_{lim}$ and for different values of $f_{sp,min}$. For the interpretation of these results it is important to note that that the size distributions measured by POLDER relate to humidified particles. Compared to dry particles, there is about a factor 2 increase in particle radius if the relative humidity is >90%[20] (which is typically the case for grid cells that contain a cloud). First of all, S increases with $f_{sp,min}$, indicating that particle sphericity is a good indicator of the capability of an aerosol to take up water and hence nucleate droplets. An optimal value for $f_{sp,min} = 0.90$ is found. Looking at the dependence on $r_{lim}$, S is independent of $r_{lim}$ from 0 to 0.025 μm consistent with laboratory results[14] showing that such small particles are not suitable as CCN. S strongly increases for $r_{lim}$ between 0.05 and 0.15 μm, and then flattens off between 0.15 and 0.20 μm. The reason for this behavior is that particles with (wet) radius > 0.15 μm are suitable as CCN even at low supersaturation (0.1%), while the aerosol burden including smaller particles may consist of a substantial fraction of non-CCN. The inclusion of non-CCN leads to an underestimate of susceptibility because variations in non-CCN have no effect on $N_d$. For $r_{lim} > 0.20$ μm, S decreases again but the determination of S becomes increasingly uncertain in this range (because of the small range in $N_{ccn}$, dominated by low values). Furthermore, for these large values of $r_{lim}$, the number of larger aerosols is no longer a good proxy for the number of aerosols at intermediate sizes that make up the bulk of the CCN. The value for $r_{lim} = 0.15$ μm corresponds to the best CCN proxy, because for this value is is expected that only CCN-capable aerosols are included in the aerosol burden. It should be noted that the actual CCN size distribution also contains smaller particles (this fraction becomes larger with increasing supersaturation) and the assumption is that relative variations in the POLDER CCN proxy $N_{ccn}$ are representative for the variations in CCN.

**Comparison between CCN proxies.** Figure 2 demonstrates that there is not a simple scaling between AOD and $N_{ccn}$. For example, areas with anthropogenic pollution around Asia, Europe, and the

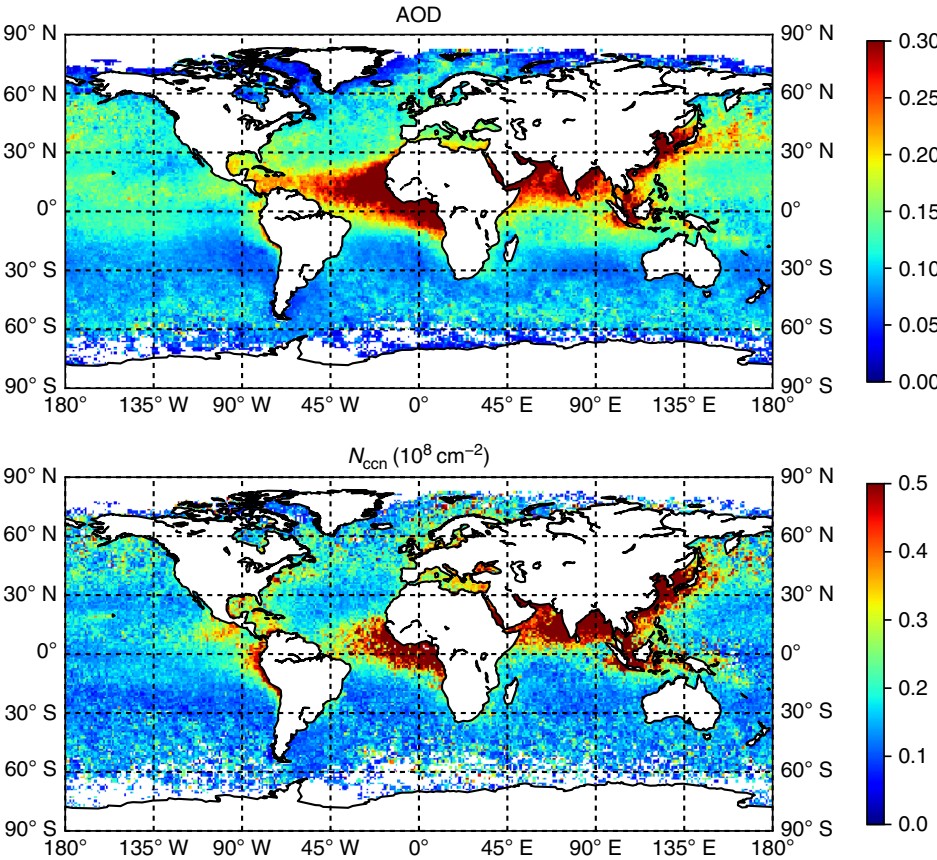

**Fig. 2** Aerosol Optical Depth and CCN column number. Annual average values of the aerosol optical depth (AOD) and cloud condensation nuclei (CCN) column number ($N_{ccn}$) for 2006.

US East coast as well as biomass burning west of Africa are more pronounced in $N_{ccn}$, whereas mineral dust transport from the Sahara over the Atlantic ocean gives a large signal in AOD but is almost absent in $N_{ccn}$.

Figure 3 shows the relationships of $N_d$ with AOD, AI, and $N_{ccn}$, respectively, for global ocean retrievals. Comparing the dependence of $N_d$ on AOD and AI, we see that for AI the dependency is stronger for higher AI values, but at AI values <0.05 there is virtually no dependence of $N_d$ on AI. This suggests that AI is a poor CCN proxy at low AI but a better CCN proxy than AOD at higher values of AI. An explanation for the behavior at low AI is that the Angstrom Exponent can get very close to 0 (meaning that the AOD is independent of wavelength), in which case AI is no longer related to particle concentration. Furthermore, as suggested by Ma et al.[12], absolute measurement uncertainties dominate at low aerosol concentrations (and hence low-AI values) which leads to an underestimate of susceptibility. When leaving out AI values <0.05 the slope increases from 0.4 to 0.57. $N_{ccn}$ shows the strongest relationship with $N_d$. Here, increase in slope, (from 0.55 to 0.66) by leaving out the small values affected by measurement uncertainties (see Methods) is less strong than for AI. Removal of the largest 10% of data would slightly enhance the slope for $N_{ccn}$, but the increase is small (0.01) and even weaker for AI.

Susceptibilities $S_{aod}$, $S_{ai}$, and $S_{ccn}$, corresponding to AOD, AI, and $N_{ccn}$, respectively, for different geographical regions[21] are shown in Fig. 4. The values were determined using the whole range for AOD, AI, and $N_{ccn}$ ($S^{full}$, solid bars), and also when ignoring the low values ($S^{opt}$, transparent bars). As shown by our simulator (see Methods), $S_{ccn}^{opt}$ provides our best estimate for susceptibility for $N_{ccn}$, as it ignores the low values affected by

measurement uncertainties. A similar reasoning would apply for AOD and AI, but all previous studies on $RF_{aci}$ used in the IPCC 5th assessment report (AR5)[2] used the whole range of AOD or AI, so in order to compare with previous work $S_{ccn}^{full}$ and $S_{ai}^{full}$ are of relevance. For all regions, both, $S_{ccn}^{full}$ and $S_{ccn}^{opt}$ are higher than or similar to the corresponding values of $S_{ai}$ and $S_{aod}$. $S_{ai}^{full}$ is either similar to or smaller than $S_{aod}^{full}$, while $S_{ai}^{opt} > S_{aod}^{opt}$ for most regions. The value $S_{ccn}^{opt} = 0.66$ for $N_{ccn}$ for the global ocean (range between 0.4 and 0.85 for the different regions) is about 50% higher than both $S_{aod}^{full}$ (0.41) and $S_{ai}^{full}$ (0.40), which are based on the approach used in previous studies included in IPCC-AR5[2]. Looking at the different regions, the relative difference between $S_{ccn}^{opt}$ and $S_{aod}^{full}$ are largest in NAO, TAO, NPO and TPO. The relative difference between $S_{ccn}^{opt}$ and $S_{ai}^{full}$ are for most regions similar to the global difference, except for SPO, SIO, and SAO which are strongly affected by the $N_d$-AI behavior at low AI. The susceptibility we find using $N_{ccn}$ is closer to the values found by in situ studies[22] than the susceptibilities based on AOD or AI. The susceptibility for AI ($S_{ai}^{opt}$) gets much closer to that of $N_{ccn}$ ($S_{ccn}^{opt}$) when AI values <0.05 are left out.

**Radiative forcing due to aerosol–cloud interactions**. Based on $S_{ccn}^{opt}$, we determine $RF_{aci}$ using different aerosol–climate models[8] to compute the increase $\Delta N_{ccn}$ between pre-industrial times (PI) and present-day (PD). Here, we assume that our derived susceptibilities are applicable to the PI–PD increase in vertically integrated CCN concentration at 0.3% supersaturation. From $\Delta N_{ccn}$ we compute $\Delta N_d$ using the values of $S_{ccn}^{opt}$ for the different regions, the resulting change in cloud albedo using the Twomey formula[1] and the

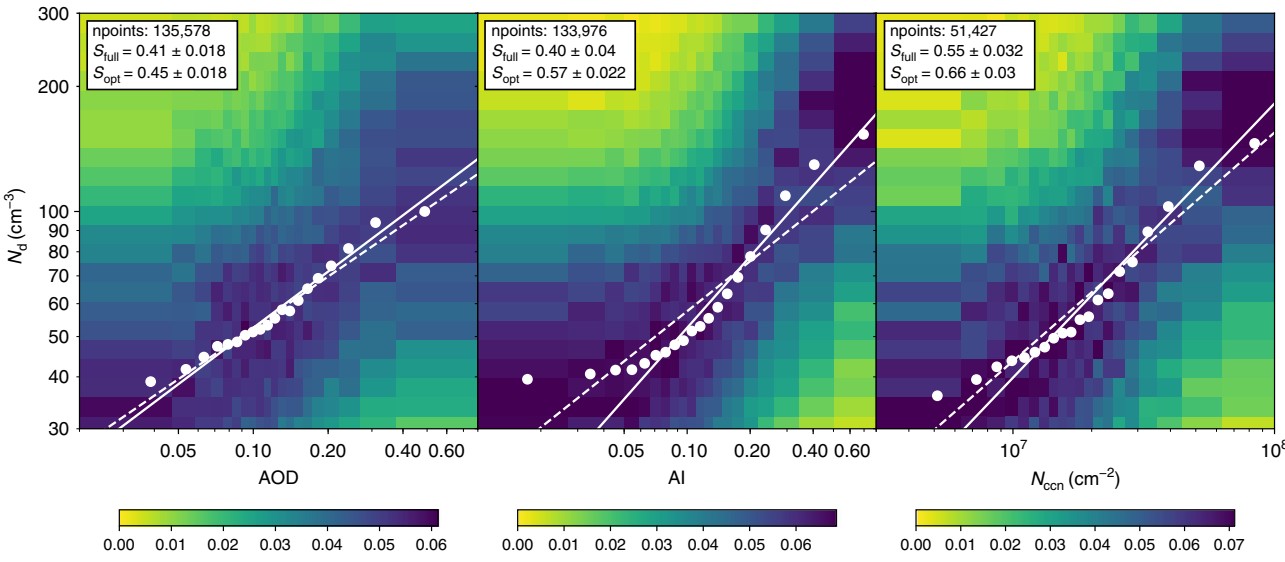

**Fig. 3** Dependence of cloud droplet number concentration on aerosol. Cloud droplet number concentration ($N_d$) versus aerosol optical depth (AOD), aerosol index (AI), and cloud condensation nuclei (CCN) column number ($N_{ccn}$), for the global ocean data set for 2006. Each point represents a bin median of $N_d$ and CCN proxy, where each bin contains the same number of points ($n$ points/20). The colors indicate the normalized histogram of $N_d$ in each AOD/AI/$N_{ccn}$-bin. The dashed lines show the linear regression through all data points and the solid lines using only data points for $N_{ccn} > 10^7 \text{cm}^{-2}$, AI > 0.05, and AOD > 0.07, leaving out the lowest 15% of data for all 3 proxies. The quoted errors are $2\sigma$ errors on the regression slope.

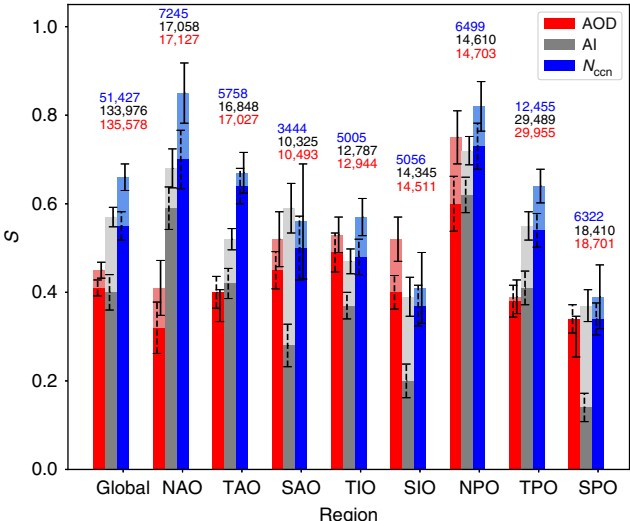

**Fig. 4** Susceptibilities for different CCN proxies. Susceptibilities based on aerosol optical depth (AOD), aerosol index (AI), and cloud condensation nuclei (CCN) column number ($N_{ccn}$) for global ocean data and for the regions defined by Quaas et al.[39]: North Pacific Ocean (NPO), Tropical Pacific Ocean (TPO), South Pacific Ocean (SPO), Southern Indian Ocean (SIO), South Atlantic Ocean (SAO), Tropical Indian Ocean (TIO), Tropical Atlantic Ocean (TAO), and North Atlantic Ocean (NAO). Solid bars are based on the full range of $N_{ccn}$, AI, and AOD values while transparent bars only use the ranges $N_{ccn} > 10^7 \text{cm}^{-2}$, AI > 0.05, and AOD > 0.07, respectively. Error bars indicate the standard error ($2\sigma$) of the linear regression and do not include the systematic error term of ($+0.06$ for $S_{ccn}^{opt}$) that follows from the simulator. The number of aerosol-$N_d$ pairs are indicated for proxy above the bars.

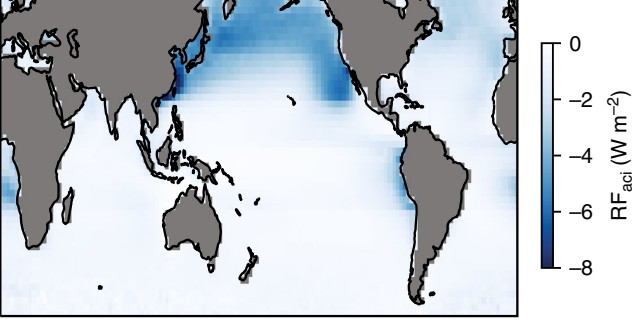

**Fig. 5** Spatial distribution of radiative forcing over ocean. Spatial distribution of the radiative forcing due to aerosol–cloud interactions over the ocean ($RF_{aci,ocean}$), based on the mean of the different model estimates for the increase in cloud condensation nuclei (CCN) column number between present day (PD) and pre-industrial (PI) times.

resulting $RF_{aci,ocean}$ (over ocean) using the approach of Gryspeerdt et al.[8]. The strongest contribution to $RF_{aci,ocean}$ comes from the northern Pacific ocean (Fig. 5). To estimate the contribution over land, we use the scaling factors between the global- and ocean-only

$RF_{aci}$ value from different aerosol climate models[23]. Combining the different values of $RF_{aci,ocean}$, from the different model estimates of $\Delta N_{ccn}$, with the different ratios $RF_{aci}/RF_{aci,ocean}$ and the expected uncertainty distribution due to uncertainty in $S_{ccn}^{opt}$ (from the simulator), we get a histogram of possible values for $RF_{aci}$ (Fig. 6). We take the median of this distribution, $-1.14 \text{ W m}^{-2}$, as our best $RF_{aci}$ estimate, and define an uncertainty range using the 5 and 95 percentile values, respectively, which yields a range between $-0.84$ and $-1.72 \text{ W m}^{-2}$. This uncertainty range is dominated by the uncertainty in $RF_{aci}/RF_{aci,ocean}$. We expect this uncertainty to decrease when improved polarimetric observations over land become available, in particluar from the NASA phytoplankton, aerosol, cloud and ocean ecosystem mission[24,25].

To test the sensitivity to the assumption that the POLDER derived susceptibilities for $N_{ccn}$ are applicable to the column CCN at 0.3% supersaturation, we also computed $RF_{aci}$ using the PI–PD increase in column CCN at 0.1% supersaturation. This yields virtually the same $RF_{aci}$ best estimate and range (within $0.01 \text{ W m}^{-2}$), which

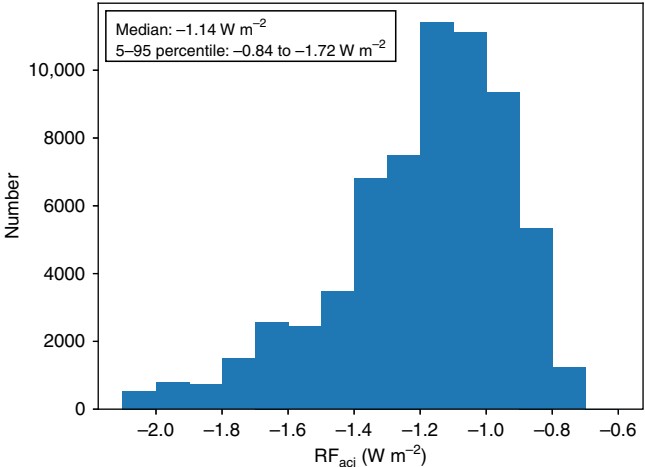

**Fig. 6** Histogram of $RF_{aci}$. Histogram of the Radiative Forcing due to aerosol-cloud interactions ($RF_{aci}$) based on the different values of $RF_{aci,ocean}$ from the different model estimates for the increase in cloud condensation nuclei (CCN) column number between present day (PD) and pre-industrial (PI) times, the different model values of the ratio $RF_{aci}/RF_{aci,ocean}$, and the effect of the uncertainty distribution of the susceptibility $S_{ccn}^{opt}$ derived in this study.

demonstrates our assumption on this aspect does not contribute to the uncertainty in $RF_{aci}$.

The values $RF_{aci,aod}$ (based on $S_{aod}^{full}$) and $RF_{aci,ai}$ (based on $S_{ai}^{full}$) are $-0.33\,\mathrm{W\,m^{-2}}$ (range: $-0.19$ to $-0.54\,\mathrm{W\,m^{-2}}$) and $-0.80\,\mathrm{W\,m^{-2}}$ (range: $-0.58$ to $-1.24\,\mathrm{W\,m^{-2}}$), respectively. Here, the differences between $RF_{aci,aod}$ and $RF_{aci,ai}$ can be explained by the fact that the PD–PI increase in AI from the models is much larger than the increase in AOD. Our ranges overlap with previous estimates based on AI and AOD[4,8] but the best estimates are more negative. This is mainly because those earlier studies assumed a very small contribution to $RF_{aci}$ over land, where satellites have poor capabilities in providing AI/AOD–$N_d$ relationships[8] (our ocean-only values for $RF_{aci,aod}$ and $RF_{aci,ai}$ are very close to the earlier studies[4,8]). It is interesting to note that if we use $S_{ai}^{opt}$ we obtain an $RF_{aci}$ estimate of $-1.04\,\mathrm{W\,m^{-2}}$, i.e., close to our best estimate. This suggests that the main issue with the use of AI as in previous studies is its behavior at small values, and in other aspects it appears to be a useful proxy for CCN[8].

## Discussion

It is not straightforward to scale our $RF_{aci}$ estimate to an effective radiative forcing due to aerosol–cloud interactions ($ERF_{aci}$), which also includes other aerosol induced changes in cloud properties (e.g., cloud fraction and liquid water path). However, recent studies[23,26] suggest that $ERF_{aci}$ is at least as negative as $RF_{aci}$ because the negative effect (in terms of radiation) of cloud fraction adjustment is most likely stronger than the positive effect of LWP adjustment. The top-down study on the Earth energy balance by Murphy et al.[27] finds a possible range (90% confidence) in total aerosol radiative forcing ($ERF_{aci+ari}$), including also aerosol-radiation interactions, between $-0.3$ and $-1.9\,\mathrm{W\,m^{-2}}$ (best estimate $-1.06\,\mathrm{W\,m^{-2}}$), which is close to the estimated $ERF_{aci+ari}$ range of Bellouin et al.[28], between $-0.4$ and $-2.0\,\mathrm{W\,m^{-2}}$ (90% confidence, which infers the lower bound of $-2.0\,\mathrm{W\,m^{-2}}$ also by a top-down approach). Taking our $RF_{aci}$ estimate as a lower bound for $ERF_{aci}$, and assuming $RF_{ari}$ due to aerosol-radiation interactions to be at least[2] $-0.2\,\mathrm{W\,m^{-2}}$, we see that our estimated $RF_{aci}$ range is plausible given the top-down estimates, although the most negative values are only realistic if $RF_{ari}$ is weak and the LWP and cloud fraction effects cancel each other out. Also, our estimate is

plausible given estimates from climate models constrained by pollution trends[29] (between $-0.90$ and $-1.70\,\mathrm{W\,m^{-2}}$ at 68% confidence).

The IPCC-AR5 estimate[2] of $ERF_{aci}$, is $-0.45\,\mathrm{W\,m^{-2}}$. This relatively weak negative forcing, which is reduced compared to IPCC-AR4[30], is a result from an expert judgement where a large weight was given to satellite based studies. It has already been suggested by previous authors[8,9], and supported by our results, that some of the earlier satellite-based estimates of $RF_{aci}$, used in IPCC-AR5, were biased low due to the use of AOD as CCN proxy. We find that also the AI estimates are biased low by almost 50% due to issues at low-AI values. Our $RF_{aci}$ estimate overcomes the known issues with previous estimates by using aerosol measurements more directly related to aerosol–cloud interactions (aerosol number, size, and shape)[14,15] and by using only measurements in the range not dominated by measurement uncertainties. The lower bound of our range ($RF_{aci} = -0.84\,\mathrm{W\,m^{-2}}$) is almost a factor 2 more negative than the IPCC-AR5 estimate of $ERF_{aci}$, and is actually more in line with the IPCC-AR4 estimate. Our best estimate of $RF_{aci} = -1.14\,\mathrm{W\,m^{-2}}$ is even more negative than the IPCC-AR5 estimate of $-0.90\,\mathrm{W\,m^{-2}}$ for the total aerosol radiative forcing ($ERF_{aci+ari}$). These findings put into question that by expert judgement the satellite studies were given more weight than model estimates in IPCC-AR5, resulting in a weaker negative forcing than IPCC-AR4, in particular because our estimate is more in line with the models and with IPCC-AR4. A stronger aerosol cooling indicates that the global temperature is more sensitive to anthropogenic greenhouse gas emissions than previously assumed[31], because it partly masks the warming by greenhouse gases[32].

## Methods

**POLDER-3 aerosol retrievals.** In this work, we use the POLDER-3 aerosol product retrieved by the SRON aerosol retrieval algorithm[16,17] (processed for the year 2006) previously used for computing the direct radiative effect of aerosols[18]. Retrievals are based on Collection-3 level-1 POLDER-3 medium resolution data ($18 \times 18\,\mathrm{km^2}$). We restrict ourselves to retrievals over ocean because of limited quality of over-land retrievals, especially for cases with low aerosol loading. Furthermore, we restrict the study to $60°S <$ latitude $<60°N$. POLDER-3 achieves global coverage in 1.5 days.

Cloud screening has been performed based on goodness-of-fit between forward model and measurement[17]. This means aerosol retrievals in (partly) cloudy scenes and directly next to clouds are excluded[33]. The aerosol products have been gridded on a $1° \times 1°$ latitude-longitude grid. The aerosol retrieval algorithm is based on a description of aerosols by a fine and a coarse mode (indicated by superscripts $f$ and $c$, respectively), where each mode is described by a log-normal function. Aerosol properties in the state vector are for both modes the effective radius $r_{eff}$, effective variance $v_{eff}$, the real and imaginary part of the refractive index $m_r$ and $m_i$, and the aerosol column number concentration $N_a$. The fraction of spheres[34] $f_{sp}$ of the coarse mode is included as a fit parameter in the retrieval state vector. In this study, we use the AOD, $N_a$, $r_{eff}$, and $v_{eff}$ of the fine and coarse mode, and $f_{sp}$.

$N_{ccn}$ is computed from the log-normal bi-modal size distribution for the retrieved $N_a$, $r_{eff}$ and $v_{eff}$ of the fine and coarse mode as the column number of particles with radius $> r_{lim} = 0.15\,\mu m$. To investigate the capability of POLDER-3 to provide $N_{ccn}$, we created 1000 synthetic POLDER-3 measurements with varying aerosol properties in the following range (superscripts $f$ and $c$ indicate fine and coarse mode, respectively): $0.02$–$0.3\,\mu m$ for $r_{eff}^f$, $0.65$–$3.5\,\mu m$ for $r_{eff}^c$, $0.1$–$0.3$ for $v_{eff}^f$, $0.4$–$0.6$ for $v_{eff}^c$, $1.33$–$1.60$ for $m_r^f$ and $m_r^c$, $10^{-8}$–$0.1$ for $m_i^f$, $10^{-8}$–$0.02$ for $m_i^c$, $0.005$–$0.7$ for $AOD^f$ and $AOD^c$, and $0$–$1$ for $f_{sp}$. We put a random error on the synthetic measurements with a standard deviation of 2% for radiance and 0.015 for degree of polarization, which is representative for POLDER-3 measurements over ocean[35]. From the synthetic experiment we conclude that the uncertainty on individual retrievals of $r_{eff}^f$ is $0.034\,\mu m$ (bias $0.016\,\mu m$) and on $f_{sphere}$ 0.25 (bias 0.13). For $N_{ccn}$ we find that for 71% of the data the difference between retrieved and true $N_{ccn}$ is smaller than $0.20 \cdot N_{ccn} + 4 \times 10^6\,\mathrm{cm^{-2}}$. We use this as an uncertainty estimate of $N_{ccn}$.

Comparing POLDER $N_{ccn}^{pol}$ with corresponding values $N_{ccn}^{aer}$ computed from the aerosol size distribution of ground-based aerosol robotic network (AERONET) measurements, we find (Supplementary Fig. 1) $R^2 = 0.58$, a bias of $8.0 \times 10^6\,\mathrm{cm^{-2}}$, a mean absolute difference of $1.95 \times 10^7\,\mathrm{cm^{-2}}$, and a root mean square difference of $3.90 \times 10^7\,\mathrm{cm^{-2}}$. Totally, 51% of the POLDER-AERONET differences are smaller than the error bound found from the synthetic

experiment ($0.20 \cdot N_{ccn} + 4 \times 10^6\,cm^{-2}$), which suggest that this is a reasonable error estimate given that the differences are also affected by errors in AERONET data. Most important, there is no significant trend of the relative difference (($(N_{ccn}^{pol} - N_{ccn}^{aer})/N_{ccn}^{pol}$) with $N_{ccn}^{pol}$ ($R^2 = 0.01$), which would affect the $N_{ccn} - N_d$ relationships.

**MODIS cloud retrievals**. We use the MODIS Collection-6 retrievals of cloud effective radius (CER) and cloud optical thickness (COT)[19] to compute the cloud droplet number concentration $N_d$ using the adiabatic approximation[11,36], and aggregate the data at a $1° \times 1°$ horizontal and daily temporal resolution. Here, we consider only points for which CER > 4 μm and COT > 4, cloud fraction > 0.9 (at 5 km resolution), and with a sub-pixel inhomogeneity index (cloud_mask_SPI) < 30, as these are the ranges where meaningful CER and COT retrievals can be performed[36]. Further, we only consider liquid clouds by selecting $1° \times 1°$ grid cells with COT_ice = 0.

**Deriving susceptibility**. To derive the susceptibility from POLDER-3 and MODIS data, we use all grid cells for the year 2006 for which we have both a valid POLDER-3 $N_{ccn}$ value and a valid MODIS $N_d$ value. After selecting the data points in the region under consideration, we define a number of bins for $N_{ccn}$ where each bin has an equal number of points. For deriving susceptibility we use 100 bins. The $N_{ccn}$ value attributed to a certain bin is the median of all $N_{ccn}$ values in that bin. The $N_d$ value attributed to this bin is the median of all $N_d$ values collocated with the $N_{ccn}$ retrievals in that bin. This procedure gives us 100 paired values of $N_{ccn}$ and $N_d$ for the region under consideration. So, for determining the global (ocean) value for susceptibility we aggregate all valid POLDER3-MODIS data pairs into 100 bins. Having the same number of measurements in each bin ensures that each bin has the same statistical representativity. The susceptibility is determined by fitting a linear regression through the binned points of $N_d$ versus $N_{ccn}$. The derived susceptibilities do not change significantly if a different number of bins is chosen (e.g., for 20 or 1000 bins the global susceptibilities agree within 0.01 with those for 100 bins).

An important assumption we make here is that the aerosol properties are uniformly distributed over a grid cell and that relative variations in the column integrated aerosol concentration are representative for the aerosol that interacts with the cloud at cloud base. Under this assumption, it is unnecessary to perform retrievals below clouds or directly next to clouds, which is not possible with current retrieval algorithms[33]. These assumptions may lead to an underestimation of susceptibility because if the retrieved aerosol is not representative for the aerosol that interacts with clouds, variations in retrieved aerosol properties would have no effect on $N_d$. Aspects related to spatial sampling by the satellite are not expected to impact the derived susceptibility[12].

**Simulating the effect of measurement errors**. To investigate the effect of measurement errors on $N_{ccn}$, we use a simulator that models $N_d$ as function of $N_{ccn}$. For the ensemble of $N_{ccn}$ we use the values measured by POLDER and we compute the corresponding $N_d$ assuming $N_d = C\,N_{ccn}^S$, with S = 0.66 and C = 0.001. When we apply our procedure to determine susceptibility (see above) to the simulated data set we find exactly S = 0.66, as expected. When we put 50% error ($1\sigma$) error on $N_d$ we still find S very close (within 0.01) to 0.66, showing that derived susceptibilities are not very sensitive to random fluctuations on $N_d$. Next, we put random errors on $N_{ccn}$ of the form $err_{ccn} = rel_{err}N_{ccn} + abs_{err}$, where $rel_{err}$ and $abs_{err}$ denote the relative- and absolute error on $N_{ccn}$, respectively. We choose a range ($1\sigma$) for $rel_{err}$ of 0–0.5 and $abs_{err}$ of $0$–$5 \times 10^6\,cm^{-2}$. From the simulated data with error we determine the susceptibility $S_{full}$ using the whole range of $N_{ccn}$ and $S_{opt}$ using only values > $10^7\,cm^{-2}$, leaving out 15% of the smallest values which are most heavily affected by the absolute term in the measurement uncertainty.

Supplementary Fig. 2 shows $S_{full}$ (left panel) and $S_{opt}$ (right panel) as function of $abs_{err}$ for different values of $rel_{err}$. It can be seen that $S_{full}$ is being under-estimated in the presence of measurement errors, as is expected from earlier results based on AOD[12]. The values for $S_{opt}$ are much closer to the true value of 0.66, although under-estimation is still possible (up to 0.18 for the range of $N_{ccn}$ errors shown). The simulation results confirm that $S_{opt}$ is a better estimate of susceptibility than $S_{full}$. The expected measurement uncertainty of $0.20 \times N_{ccn} + 4 \times 10^6\,cm^{-2}$ indicates an underestimation in $S_{opt}$ of 0.06. Choosing a larger cut-off value for $N_{ccn}$ than $10^7\,cm^{-2}$ does not reduce the uncertainty range.

**Radiative forcing calculation**. We use five different global aerosol climate models (ECHAM6-HAM, CAM5.3, CAM5.3-CLUBB, CAM5.3-CLUBB-MG2, and SPRINTARS) to compute the increase $\Delta N_{ccn}$ (using the column CCN at 0.3% supersaturation) between PI and PD from a pair of nudged simulations that are the same except that the PI simulation uses pre-industrial, and the PD simulation, PD aerosol emissions. All these models, which were also used in the study by Gryspeerdt et al.[8], have participated in the AEROCOM intercomparisons in current or previous model versions[37,38]. From $\Delta N_{ccn}$, for the different models, we compute $\Delta N_d$ using the values for S as derived from POLDER-3 and MODIS. From $\Delta N_d$ we

compute the change in cloud albedo using the Twomey formula[1] and the $RF_{aci}$ using Eq. (3) of Gryspeerdt et al.[8]

$$RF_{aci} = -F^\downarrow\, f_{liq}\, \alpha_{cld}(1 - \alpha_{cld})\frac{1}{3}\frac{\Delta N_d}{N_d}, \qquad (1)$$

where $F^\downarrow$ is the daily-mean incoming solar radiation flux at each grid-point for each day, $f_{liq}$ the fractional cover by liquid–water clouds, and $\alpha_{cld}$ the cloud albedo taken from CERES. Since we only derived values for susceptibility over ocean, the procedure above gives an estimate $RF_{aci,ocean} = -0.76\,W\,m^{-2}$ over the ocean and a range between $-0.68$ and $-0.85\,W\,m^{-2}$ from the range in $\Delta N_{ccn}$ from the different models. The procedure for computing $RF_{aci,aod}$ and $RF_{aci,ai}$ is the same as for $N_{ccn}$ except that they are based on the PD–PI increase in AOD and AI, respectively.

To get an estimate of the land contribution to $RF_{aci}$ we looked at the ratio $RF_{aci}/RF_{aci,ocean}$ in 13 different aerosol climate models[23]. This gives us a range of values for $RF_{aci}/RF_{aci,ocean}$ between 1.12 and 2.24, and a mean value of 1.5. Further, the simulator results indicate that for the expected error on $N_{ccn}$ ($0.20 \times N_{ccn} + 4 \times 10^6\,cm^{-2}$) the global value of $S_{ccn}^{opt}$ is underestimated by 10%, but given that this error estimate itself is quite uncertain also smaller and larger underestimations cannot be ruled out. Therefore, we model this uncertainty with a normal distribution of the $RF_{aci}$ scaling factor with mean 1.05 and standard deviation 0.05, assuming a linear dependence of $RF_{aci}$ on susceptibility. Combining this distribution with the different values for $RF_{aci,ocean}$ and the different values of the ratio $RF_{aci}/RF_{aci,ocean}$, we get a histogram of possible values for $RF_{aci}$ We take the median of this this distribution, $-1.14\,W\,m^{-2}$, as our best $RF_{aci}$ estimate, and define an uncertainty range using the 5 and 95 percentile values, respectively, which yield a range between $-0.84$ and $-1.72\,W\,m^{-2}$. If we ignore the uncertainty on $S_{ccn}^{opt}$ we obtain a range for $RF_{aci}$ between $-0.80$ and $-1.60\,W\,m^{-2}$ and if we assume an error on $S_{ccn}^{opt}$ that is twice as large (mean 1.10 and standard deviation 0.10) we obtain a range between $-0.86$ and $-1.84\,W\,m^{-2}$. If we would only take into account the uncertainty in the ratio $RF_{aci}/RF_{aci,ocean}$, the resulting $RF_{aci}$ range would be from $-0.85$ to $-1.70\,W\,m^{-2}$. So, by far the largest part of the total uncertainty range can be explained by the uncertainty in the ratio $RF_{aci}/RF_{aci,ocean}$ and the uncertainty caused by the uncertainty on $S_{ccn}^{opt}$ is small compared to the total uncertainty.

When computing the uncertainty ranges for $RF_{aci,aod}$ (based on $S_{aod}^{full}$) and $RF_{aci,ai}$ (based on $S_{ai}^{full}$) we do not take the uncertainty in S into account, in order to make these estimate comparable to previous studies.

## Data availability
The data set analyzed during the current study is available through ftp.sron.nl/open-access-data/hasekamp/NatureComm2019/. The AEROCOM model history is available at aerocom.met.no/data.html.

## Code availability
The computer codes used to analyze the data are available from the corresponding author on reasonable request.

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

## Acknowledgements

POLDER/PARASOL Level-1 data originally provided by CNES and distributed by ICARE/LOA/LSCE. MODIS data were provided by the NASA Goddard Space Flight Centre and CERES data from the NASA Langley Research Centre. The authors would like to thank Andrew Gettelman and Hugh Morrison (NCAR), David Neubauer (ETH-Zurich), Toshihiko Takemura (Kyushu University), Hailong Wang and Kai Zhang (PNNL) and Minghuai Wang (Nanjing University), for performing the models simulations used in this work as part of the AeroCom Initiative. The output from these simulations is available from aerocom.met.no upon request. EG is supported by an Imperial College Junior Research Fellowship. The work of J.Q. is supported by the European Union via its Horizon 2020 projects CONSTRAIN (GA 820829) and FORCeS (GA 821205). The collaboration for this study resulted from the international team on "Study of aerosol–cloud interactions based on satellite observations of the terrestrial underlying surface–atmosphere system: a new frontier of atmospheric science", lead by A. Kokhanovsky and D. Rosenfeld and hosted by the International Space Science Institute (ISSI).

## Author contributions

O.H. and J.Q. designed the research, O.H. and E.G. performed the research, O.H. and J.Q. wrote the paper.

## Competing interests

The authors declare no competing interests.
