## [Peer Review File · Nature Communications]

Reviewers' comments:

Reviewer #1 (Remarks to the Author):

The authors have properly addressed my opinions in my first review report. I have only one more comment with the revised paper. When the authors further investigated the uncertainty level to which POLDER can constrain the estimate of Nccn, they used 2% radiance error and 0.015 degree of linear polarization in retrieval simulation. I suggest adding reference papers for these numbers - viewing a) the strong impact of instrument error propagation to the retrievals and b) some existing diversity in accounting for POLDER measurements. In addition, are these errors accounted as random noise or systematic errors in the simulation ?

Reviewer #2 (Remarks to the Author):

This paper uses aerosol retrievals from a satellite-based polarimeter instrument to estimate the number of CCN (Nccn) over global oceans. Collocated estimates of cloud droplet concentrations (Nd) are also made using MODIS. Linear regression between the aerosol metric and Nd are performed to determine the slope of the relationship (termed the susceptibility, S). The same thing is done with other metrics of aerosol (AOD and AI) that have been used in the literature in the past. Larger values of S are found when using Nccn, which give larger estimates of cloud albedo forcing when combined with the change in aerosol from pre-industrial to present day predicted by AeroCom models. It is also found that excluding the lower range of Nccn, AI, or AOD leads to higher S values that are likely to be closer to the real values.

It is clear that polarimetry retrievals of aerosol are more sophisticated and provide more information than those from satellites such as MODIS and that they can be used to obtain a better proxy for the number of CCN. Therefore the use of such retrievals is novel and exciting. It is also useful to learn that excluding the lower AI (and AOD/Nccn) values, which are likely to be more uncertain, leads to higher S values and that these are likely closer to truth according to the error simulator; this indicates that the extra information from POLDER may not be absolutely necessary to obtain reasonable S estimates since more traditional AI techniques may be adequate. The work builds upon that of Ma (2018; <https://www.nature.com/articles/s41467-018-05028-4>), which focused only on AOD based estimates of S and will hopefully have some influence on the next IPCC report (along with the Ma study), so that less weight is put on direct estimates of S from satellites given the likely inaccuracies that would occur if using such values directly to calculate a forcing. The authors have addressed the comments from my previous review and that from the other referees well and as such I think it now deserves publication in Nature Communications once the few more comments below have been addressed.

Specific comments

In the response to the reviewer comments you mentioned that you "include the range in Sopt from the simulator in the uncertainty range on RFaci". This should be mentioned in the methods (and perhaps the caption of Fig. 4 too. Presumably this is for Nccn errors following the relationship $0.2N_{ccn} + 4 \times 10^6$? This should also be mentioned. It would also be useful to quantify what the effect on the RF uncertainty of larger assumed errors are (i.e., the other relationships shown in Fig. 2 of the extended data) since the estimate of the uncertainty is itself uncertain.

Perhaps a bit more discussion of the Ma (2018) in the introduction and discussion, etc. is warranted I think since it is quite relevant to this study. For example, regarding the error simulator (Fig. 2 of the extended data) – the Ma study provides an explanation for the finding that the inclusion of the low Nccn/AOD/AI values leads to underestimates in S.

Fig. 4 error bars – is 1-sigma for the fit errors sufficient? Given that it has been shown that the accuracy of the fits is of major importance here, and that the relationships don't necessarily appear to be linear, perhaps 2-sigma error bars would be more appropriate? Also, you should also include the errors calculated for Fig. 2 of the extended data (i.e., the estimated effect of the measurement uncertainty).

L64 – “The reason for this behavior is that particles with (wet) radius $> 0.15 \mu\text{m}$ are suitable as CCN even at low supersaturation (0.1%)”

- I'm not sure how this explains the lower susceptibilities? Could it not be that the number of larger aerosols is not necessarily a good proxy for the number of aerosol at intermediate sizes that make up the bulk of the CCN/droplets?

L127- “The values $R_{\text{Faci};\text{aod}}$ (based on $S_{\text{full aod}}$) and $R_{\text{Faci};\text{ai}}$ (based on $S_{\text{full ai}}$) are $-0.33\text{W}/\text{m}^2$ (range -0.19 to $-0.54 \text{W}/\text{m}^2$) and $-0.80 \text{W}/\text{m}^2$ (range -0.58 to $-1.24 \text{W}/\text{m}^2$), respectively.”

- The large difference between the RF estimates for AOD and AI are a little curious given that the global S value for AOD from Fig. 4 is actually larger than that for AI. Is it to do with different regions having different amounts of influence. It would be good to mention this along with an explanation. Also, it would also be good to quantify which regions are the most influential in terms of the contribution to the RF somewhere.

- Presumably the same change in the number of CCN at 0.3% that is used for the RF estimates for N_{ccn} is also used for the AI and AOD estimates? I.e., there is no estimate of the PI to PD change in AOD and AI? (And so this cannot be an explanation for the above?). This should be made clear in the methods.

L140 – “Top-down studies on the Earth energy balance²⁷ find a possible range in total aerosol radiative forcing ($ER_{\text{Faci}+\text{ari}}$), including also aerosol-radiation interactions, between -0.3 and $-1.9 \text{W}/\text{m}^2$ ”

- How well accepted is this top-down estimate range? Has the work been replicated / improved more recently since 2009? Or has subsequent work cast doubt on the findings? Presumably there is likely some uncertainty in this that might allow a larger aerosol forcing? Since if the upper estimate for aerosol forcing was well known then it seems this would have placed fairly large constraints on research into aerosol forcing since then (both model and observational estimates)? The citing of more than one paper for this should be considered I think given its importance.

L243 – “although under-estimation is still possible (up to 0.12).”

- Looks like the underestimate is even larger than 0.12 in some cases (up to around 0.18). Does this need qualifying for a particular combination of relative and absolute errors?

Typos

L37 – “Also studies using AI find susceptibilities that are substantially smaller than what models predict¹⁴.”

- This sentence doesn't quite work. Something like this sounds better I think: “However, studies using AI also find susceptibilities that are substantially smaller than what models predict¹⁴.”

L51 – “as linear regression coefficient” -> “as a linear regression coefficient”

L66 – “fraction on non-CCN” -> “fraction of non-CCN”

L67 – “leads an underestimate” -> “leads to an underestimate”

L118 – “We take the median of this this”

L126 – “uncertainty on R_{Faci} ” -> “uncertainty in R_{Faci} ”

L137 – “that includes also” -> “which also include”

L145 – “realistic for if R_{Fari} ” -> “realistic if R_{Fari} ”

Reviewer #3 (Remarks to the Author):

The authors have thoughtfully addressed the concerns of the previous review and I am fine with accepting the current manuscript as is.

Response to reviews of manuscript "New satellite analysis suggest stronger aerosol cooling due to aerosol-cloud interactions" by Hasekamp, Gryspeerdt, Quaas.

We would like to thank the reviewers for their comments on the revised manuscript.

Below, we provide a point-to-point response to the reviewer comments and corresponding revisions. In our response, we refer to the pages of the 'Track Changes' (TC) version of the manuscript.

We have created a separate pdf file with supplementary figures but for convenience of the evaluation process we have also included these in the main article file.

Reviewers' comments:

Reviewer #1 (Remarks to the Author):

The authors have properly addressed my opinions in my first review report. I have only one more comment with the revised paper. When the authors further investigated the uncertainty level to which POLDER can constrain the estimate of N_{ccn} , they used 2% radiance error and 0.015 degree of linear polarization in retrieval simulation. I suggest adding reference papers for these numbers - viewing a) the strong impact of instrument error propagation to the retrievals and b) some existing diversity in accounting for POLDER measurements. In addition, are these errors accounted as random noise or systematic errors in the simulation?

We added a reference to Fougnie et al. (2008) and explicitly mention that these errors are used as random errors. We would like to stress however that our uncertainty estimate for N_{ccn} does not only depend on the synthetic experiment but also on the comparison to AERONET.

Reviewer #2 (Remarks to the Author):

This paper uses aerosol retrievals from a satellite-based polarimeter instrument to estimate the number of CCN (N_{ccn}) over global oceans. Collocated estimates of cloud droplet concentrations (N_d) are also made using MODIS. Linear regression between the aerosol metric and N_d are performed to determine the slope of the relationship (termed the susceptibility, S). The same thing is done with other metrics of aerosol (AOD and AI) that have been used in the literature in the past. Larger values of S are found when using N_{ccn} , which give larger estimates of cloud albedo forcing when combined with the change in aerosol from pre-industrial to present day predicted by AeroCom models. It is also found that excluding the lower range of N_{ccn} , AI, or AOD leads to higher S values that

are likely to be closer to the real values.

It is clear that polarimetry retrievals of aerosol are more sophisticated and provide more information than those from satellites such as MODIS and that they can be used to obtain a better proxy for the number of CCN. Therefore the use of such retrievals is novel and exciting. It is also useful to learn that excluding the lower AI (and AOD/Nccn) values, which are likely to be more uncertain, leads to higher S values and that these are likely closer to truth according to the error simulator; this indicates that the extra information from POLDER may not be absolutely necessary to obtain reasonable S estimates since more traditional AI techniques may be adequate. The work builds upon that of Ma (2018; <https://www.nature.com/articles/s41467-018-05028-4>), which focused only on AOD based estimates of S and will hopefully have some influence on the next IPCC report (along with the Ma study), so that less weight is put on direct estimates of S from satellites given the likely inaccuracies that would occur if using such values directly to calculate a forcing. The authors have addressed the comments from my previous review and that from the other referees well and as such I think it now deserves publication in Nature Communications once the few more comments below have been addressed.

Specific comments

In the response to the reviewer comments you mentioned that you “include the range in S_{opt} from the simulator in the uncertainty range on R_{Faci}”. This should be mentioned in the methods (and perhaps the caption of Fig. 4 too. Presumably this is for Nccn errors following the relationship $0.2N_{ccn} + 4 \times 10^6$? This should also be mentioned. It would also be useful to quantify what the effect on the R_F uncertainty of larger assumed errors are (i.e., the other relationships shown in Fig. 2 of the extended data) since the estimate of the uncertainty is itself uncertain.

In the 'Methods' section of the revised manuscript (p15 of the TC version) we mention that the assumed error on S_{opt} is based on the line for $0.2N_{ccn} + 4 \times 10^6$ in Supplementary Figure 2. Furthermore, we include a discussion on the effect on the R_{Faci} range if we take an error on S_{opt} that is twice as large and also if we ignore this error term (p15/16 of the TC version).

Perhaps a bit more discussion of the Ma (2018) in the introduction and discussion, etc. is warranted I think since it is quite relevant to this study. For example, regarding the error simulator (Fig. 2 of the extended data) – the Ma study provides an explanation for the finding that the inclusion of the low Nccn/AOD/AI values leads to underestimates in S.

At the end of the Introduction of the revised manuscript, we refer to the study of Ma et al. with the statement: " In addition to the use of non-optimal CCN proxies, also measurement uncertainties, especially at low aerosol concentrations, lead to an underestimate of susceptibility¹²." (REF 12 is the Ma et al paper).

Further, when discussing the AI-Nd relationship, we added (p5 of the TC version) in the revised manuscript: "Furthermore, as suggested by Ma et al.¹², absolute measurement uncertainties dominate at low aerosol concentrations (and hence low AI values) which leads to an underestimate of susceptibility"

Fig. 4 error bars – is 1-sigma for the fit errors sufficient? Given that it has been shown that the accuracy of the fits is of major importance here, and that the relationships don't necessarily appear to be linear, perhaps 2-sigma error bars would be more appropriate? Also, you should also include the errors calculated for Fig. 2 of the extended data (i.e., the estimated effect of the measurement uncertainty).

- In the revised manuscript, we use 2-sigma error bars both in Figs. 4 (and changed to 2-sigma for Figs 1 and 3 as well). For the error from the instrument simulator (Supplementary Figure 2), this systematic error term in itself is quite uncertain (as noted also by the reviewer) and we only have a global estimate that does not depend on region. Combining it with the random error term from the linear regression would hide regional differences of the error bar. Given that one of the purposes of Fig. 4 is to reflect (part) of this error term (by showing both Sfull and Sopt) and also the error due to the use of non-optimal CCN proxies (by comparing to estimates for AOD and AI), we prefer to mention this systematic error term explicitly for Sopt (+0.06) in the caption of Fig. 4, rather than combining it with the error term in the linear regression.

L64 – “The reason for this behavior is that particles with (wet) radius > 0.15 um are suitable as CCN even at low supersaturation (0.1 %)”

- I'm not sure how this explains the lower susceptibilities? Could it not be that the number of larger aerosols is not necessarily a good proxy for the number of aerosol at intermediate sizes that make up the bulk of the CCN/droplets?

We agree that would be a possible explanation for the behavior at larger values of r_lim. We add on p4 (TC version) of the revised manuscript: " Furthermore, for these large values of r_lim, the number of larger aerosols is no longer a good proxy for the number of aerosols at intermediate sizes that make up the bulk of the CCN."

L127- “The values RFaci;aod (based on Sfull aod) and RFaci;ai (based on Sfull

ai) are -0.33 W/m^2 (range -0.19 to -0.54 W/m^2) and -0.80 W/m^2 (range -0.58 to -1.24 W/m^2), respectively.”

- The large difference between the RF estimates for AOD and AI are a little curious given that the global S value for AOD from Fig. 4 is actually larger than that for AI. Is it to do with different regions having different amounts of influence. It would be good to mention this along with an explanation. Also, it would also be good to quantify which regions are the most influential in terms of the contribution to the RF somewhere.

The reason for the difference in RF_{ai} and RF_{aod} is that the PD-PI increase in AOD is much less than the increase in AI. We include this explanation in the revised manuscript (p7/8 of the TC version): " Here, the differences between RF_{aod} and RF_{ai} can be explained by the fact that the PD-PI increase in AI from the models is much larger than the increase in AOD."

Also, we add a figure in the revised manuscript (Figure 5) that shows the spatial distribution of RF_{ai} .

- Presumably the same change in the number of CCN at 0.3% that is used for the RF estimates for Nccn is also used for the AI and AOD estimates? I.e., there is no estimate of the PI to PD change in AOD and AI? (And so this cannot be an explanation for the above?). This should be made clear in the methods.

See above: The RF_{ai} and RF_{aod} estimates are based on PD-PI changes in AI and AOD, respectively. We now explicitly mention this in the 'Methods' section of the revised manuscript (p15 of the TC version): "The procedure for computing RF_{aod} and RF_{ai} is the same as for Nccn except that they are based on the PD-PI increase in AOD and AI, respectively."

L140 – “Top-down studies on the Earth energy balance²⁷ find a possible range in total aerosol radiative forcing ($\text{ER}_{\text{Faci}+\text{ari}}$), including also aerosol-radiation interactions, between -0.3 and -1.9 W/m^2 ”

- How well accepted is this top-down estimate range? Has the work been replicated / improved more recently since 2009? Or has subsequent work cast doubt on the findings? Presumably there is likely some uncertainty in this that might allow a larger aerosol forcing? Since if the upper estimate for aerosol forcing was well known then it seems this would have placed fairly large constraints on research into aerosol forcing since then (both model and observational estimates)? The citing of more than one paper for this should be considered I think given its importance.

We added a reference to a paper of Bellouin et al. (Rev. Geophys., in press) which comes to an between -0.4 and -2.0 W/m^2 (where the -2.0 W/m^2 bound is inferred from a top-down approach), i.e. close to the range of Murphy et al. Also, we quote

the range found by Cherian et al. (2014) using climate models constrained by pollution trends, which is also consistent with this range.

L243 – “although under-estimation is still possible (up to 0.12).”

- Looks like the underestimate is even larger than 0.12 in some cases (up to around 0.18). Does this need qualifying for a particular combination of relative and absolute errors?

We changed "(up to 0.12)" to "(up to 0.18 for the range of Nccn errors shown)".

The next sentence however indicates that the largest underestimates are unlikely given the expected error on Nccn of $0.20N_{ccn} + 4.10^6$.

Typos

L37 – “Also studies using AI find susceptibilities that are substantially smaller than what models predict14.”

- This sentence doesn’t quite work. Something like this sounds better I think: “However, studies using AI also find susceptibilities that are substantially smaller than what models predict14”.

Corrected

L51 – “as linear regression coefficient” -> “as a linear regression coefficient”

Corrected

L66 – “fraction on non-CCN” -> “fraction of non-CCN”

Corrected

L67 – “leads an underestimate” -> “leads to an underestimate”

Corrected

L118 – “We take the median of this this”

Corrected

L126 – “uncertainty on RFaci” -> “uncertainty in RFaci”

Corrected

L137 – “that includes also” -> “which also include”

Corrected

L145 – “realistic for if RFari” -> “realistic if RFari”

Corrected

Reviewer #3 (Remarks to the Author):

The authors have thoughtfully addressed the concerns of the previous review and I am fine with accepting the current manuscript as is.

REVIEWERS' COMMENTS:

Reviewer #2 (Remarks to the Author):

I'm happy with the revised manuscript. Except maybe the error bars on Fig. 4 should have "ends" on them and be different colours for the full and restricted range values to allow them to be differentiated when they overlap.

Response to reviews of manuscript "New satellite analysis suggest stronger aerosol cooling due to aerosol-cloud interactions" by Hasekamp, Gryspeerdt, Quaas.

We would like to thank the reviewers for their comments on the revised manuscript.

Reviewers' comments:

Reviewer #2 (Remarks to the Author):

I'm happy with the revised manuscript. Except maybe the error bars on Fig. 4 should have "ends" on them and be different colours for the full and restricted range values to allow them to be differentiated when they overlap.

We changed the error bars in Fig.4 to be dashed and solid for the full and restricted range values, respectively, and gave the error bars 'caps'.